# Celecoxib Prevents Doxorubicin-Induced Multidrug Resistance in Canine and Mouse Lymphoma Cell Lines

**DOI:** 10.3390/cancers12051117

**Published:** 2020-04-29

**Authors:** Edina Karai, Kornélia Szebényi, Tímea Windt, Sára Fehér, Eszter Szendi, Valéria Dékay, Péter Vajdovich, Gergely Szakács, András Füredi

**Affiliations:** 1Institute of Enzymology, Research Centre of Natural Sciences, Eötvös Loránd Research Network, Magyar Tudósok körútja 2, H-1117 Budapest, Hungary; karaiedina@gmail.com (E.K.); ks863@cam.ac.uk (K.S.); windt.timea@gmail.com (T.W.); 2Department of Clinical Pathology and Oncology, University of Veterinary Medicine Budapest, István utca 2, H-1078 Budapest, Hungary; sara.feher25@gmail.com (S.F.); Szendi.Eszter@univet.hu (E.S.); dr.valeria.dekay@gmail.com (V.D.); Vajdovich.Peter@univet.hu (P.V.); 3Institute of Cancer Research, Medical University of Vienna, Borschkegasse 8A, A-1090 Vienna, Austria

**Keywords:** lymphoma, multidrug resistance, P-glycoprotein, drug holiday, COX-2 inhibitors, celecoxib

## Abstract

Background: Treatment of malignancies is still a major challenge in human and canine cancer, mostly due to the emergence of multidrug resistance (MDR). One of the main contributors of MDR is the overexpression P-glycoprotein (Pgp), which recognizes and extrudes various chemotherapeutics from cancer cells. Methods: To study mechanisms underlying the development of drug resistance, we established an in vitro treatment protocol to rapidly induce Pgp-mediated MDR in cancer cells. Based on a clinical observation showing that a 33-day-long, unplanned drug holiday can reverse the MDR phenotype of a canine diffuse large B-cell lymphoma patient, our aim was to use the established assay to prevent the emergence of drug resistance in the early stages of treatment. Results: We showed that an in vitro drug holiday results in the decrease of Pgp expression in MDR cell lines. Surprisingly, celecoxib, a known COX-2 inhibitor, prevented the emergence of drug-induced MDR in murine and canine lymphoma cell lines. Conclusions: Our findings suggest that celecoxib could significantly improve the efficiency of chemotherapy by preventing the development of MDR in B-cell lymphoma.

## 1. Introduction

Despite newly developed therapies and protocols, treatment of lymphomas mostly results in transient remission because of the rapid emergence of therapy resistance [1]. Dogs have shared a common environment with humans for thousands of years. Canine lymphomas and human non-Hodgkin lymphomas (NHLs) show strong similarities in genetics, therapy response, histopathology, and disease progression [1,2,3,4,5]. Spontaneously occurring canine lymphoma has been considered a comparative animal model to study mechanisms underlying therapy resistance and to investigate novel therapeutic agents for human NHLs [6,7].

Canine lymphoma is usually treated with multiagent chemotherapy, such as the CHOP-protocol (cyclophosphamide (C), doxorubicin (H, hydroxydaunorubicin), vincristine (O, Oncovin), and prednisone (P)). Although this treatment regimen initially seems successful, tumor recurrence can be expected and the relapsed tumor is often resistant to additional treatment. Cancer cells can become therapy-resistant through multiple mechanisms that include changes in drug metabolism, the decreased uptake of chemotherapy agents, increased DNA repair, and inhibition of apoptotic pathways [1,8]. One of the most common mechanisms is based on the active efflux of drugs from the cells mediated by transport proteins of the ATP (adenosine triphosphate)-binding cassette (ABC) superfamily, including P-glycoprotein (Pgp, ABCB1 (ATP-Binding Cassette Subfamily B Member 1)), MRP1 (Multidrug Resistance-Associated Protein 1) (ABCC1 (ATP Binding Cassette Subfamily C Member 1)), and BCRP (Breast Cancer Resistance Protein) (ABCG2 (ATP-binding Cassette Subfamily G Member 2)). Pgp extrudes a wide spectrum of cytotoxic compounds from the cell, resulting in decreased intracellular drug concentrations [8,9,10]. Several classes of anticancer drugs used in human and veterinary medicine are substrates for Pgp [11]. Studies have repeatedly linked the overexpression of P-glycoprotein to drug-resistant canine cancers [8,12,13]. Expression of Pgp was demonstrated in mast cell tumors [14], mammary tumors [15], pulmonary carcinoma [16], and lymphoma [17,18]. Recently, we determined the impact of Pgp function on therapy response and survival in canine lymphoma, using a quantitative measure based on the Pgp-mediated efflux of calcein AM (acetoxymethyl) [19]. In agreement with studies measuring the Pgp-mediated efflux of rhodamine 123 [20], our results confirmed the negative effect of P-glycoprotein function on therapy response and survival [21]. 

Drug holidays have been proposed to limit the development of resistance in cancer treatment [22]. According to this concept, resistance can be reversed by introducing a pause in the treatment. In absence of chemotherapy, adaptations driving resistance often prove to be a disadvantage, providing a new selection pressure against resistant populations. The molecular basis of the negative selection during drug holidays is not known. Reversal of therapy resistance is believed to be linked to reversible, nonmutational changes. However, the required length of drug holiday in different cancer types is not precisely established because only a few clinical studies are available [22,23]. In leukemia cells, Pgp is rapidly induced by vincristine treatment, but after drug removal the expression of Pgp was shown to return to initial levels [24]. This reversibility indicates a regulatory mechanism that can switch the expression of MDR (multidrug resistance) proteins on and off during periods of treatment and drug holidays, respectively. Induction and silencing of Pgp expression may be explained by epigenetic changes and/or transcription factor activation. Interestingly, drug resistance can be modulated by the combination of epigenetic inhibitors such as histone deacetylase inhibitors (HDACis) with chemotherapy [6,25,26,27,28,29]. Additionally, cyclooxygenase (COX)-2, a key enzyme responsible for the synthesis of prostaglandins from arachidonic acid, was shown to influence Pgp expression. Cells virally transduced with COX-2 show high levels of Pgp expression and rhodamine efflux activity, however, the mechanistic link between COX-2 and Pgp remains to be identified [30]. 

To study mechanisms underlying the development of drug resistance and the effect of drug holidays, we established an in vitro treatment protocol to rapidly induce Pgp-mediated MDR in cancer cells. Our ultimate goal was to identify clinically relevant COX-2 and HDACis that may prevent or delay the emergence of drug resistance during chemotherapy. Here we show that celecoxib, a potent COX-2 inhibitor, prevents doxorubicin-induced MDR by inhibiting the expression of Pgp. Our findings suggest that inclusion of celecoxib in the treatment protocols may significantly improve efficiency of chemotherapy by preventing the development of MDR in B-cell lymphoma.

## 2. Results

### 2.1. Pgp-Mediated Drug Resistance Is Temporarily Reversed by a Drug Holiday: A Clinical Case Study of Two Canine Lymphoma Patients

Two canine patients were diagnosed with B-cell lymphoma (Appendix A). Both patients were treated according to the modified CHOP protocol (Appendix A). Multidrug resistance was quantitated by the calcein assay and the activity of Pgp was expressed as a dimensionless value (multidrug resistance activity factor (MAF)) [31].

Case 1 (Figure 1a): The calcein assay performed at the time of diagnosis indicated a lack of P-glycoprotein activity (MAF = 0.01) in the treatment-naïve tumor cells of Patient 1. After 7 rounds of treatment, the MAF value was 0.16 (day 51), which increased with an additional 4 rounds of treatment, reaching MAF = 0.24 at day 93. Pgp function, characterized by repeated calcein assays performed on isolated tumor cells, showed further increase at day 184, after 9 additional rounds of chemotherapy, reaching high levels (MAF = 0.56) usually observed in Pgp-overexpressing cell lines, such as HCT-15, CHO K1, and HCT-8, exhibiting a MAF of 0.63, 0.64, and 0.46, respectively [32]. The immunophenotype of the lymphoma remained identical during the course of the 184 days (Figure 1a).

Case 2 (Figure 1b): Tumor cells isolated from Patient 2 showed significant Pgp activity at the time of diagnosis (MAF = 0.35), which further increased following 7 additional treatments (MAF = 0.52 at day 135). At this point, due to financial reasons, the treatment was temporarily halted for 33 days. Surprisingly, this unplanned drug holiday resulted in a significant decrease of Pgp activity (MAF = 0.22). However, following two additional cycles of treatment, tumor cells regained Pgp expression (MAF = 0.31), indicating the re-emergence of the resistant phenotype at day 217. The immunophenotype of the lymphoma remained identical during the course of the 217 days (Figure 1b). 

### 2.2. A Novel Assay to Study the Rapid Emergence of Drug Resistance and the Effect of Drug Holiday

To study mechanisms underlying chemotherapy-induced drug resistance and renewed sensitivity following drug holidays, we established an in vitro assay based on the treatment of lymphoma cell lines. Our aim was to model clinical protocols by treating drug-naive cells with high concentrations of doxorubicin for 5 days, followed by culturing cells in drug-free medium. Treatment with IC_10_ (Inhibitory Concentration 10) of doxorubicin equals the concentration which kills 90% of the cells, however, surviving cells were able to recover and proliferate over time. After 3 cycles of doxorubicin (13 nM) treatment, P388 cells showed a significant increase in P-glycoprotein function (P388 D), evidenced by a dramatic increase of the MAF value (from 0.02 ± 0.02 to 0.68 ± 0.16). Intriguingly, a 32-day-long “drug holiday”, during which P388 D cells were cultured without doxorubicin (P388 D/DH), resulted in a decrease of the MAF value (0.47) (Figure 2a). In line with the observed changes in P-glycoprotein activity, P388 D cells were 9.9-fold resistant to doxorubicin (DOX) (*p* < 0.0001), while P388 D/DH cells showed only 3.6-fold resistance to DOX as compared to parental P388 cells (Figure 2b). The direct involvement of Pgp in the acquired resistance of P388 D cells was verified by the addition of the P-glycoprotein inhibitor tariquidar (TQ). In agreement with the functional results, mRNA (messenger ribonucleic acid) expression of the mouse Abcb1a gene increased in P388 D (*p* < 0.0001) and decreased in P388 D/DH cells (*p* = 0.0003), while the expression of Abcb1b was equally high in both treatment groups compared to P388 (*p* < 0.0001) (Figure 2c, Appendix A).

Similar results were obtained with a canine B-cell lymphoma cell line: Parental CLBL-1 cells express low levels of Pgp (MAF = 0.16 ± 0.03), which were significantly increased after 6 rounds of doxorubicin treatment (MAF = 0.39 ± 0.05), resulting in the increased doxorubicin resistance of the cells. Again, culturing of the cells for 27 days without doxorubicin decreased the MAF value to 0.3 (±0.04) and increased the sensitivity of cells to doxorubicin (*p* = 0.0006) (Figure 3a,b).

### 2.3. Celecoxib Prevents the Development of Pgp-Mediated Drug Resistance In Vitro

As drug holidays are not routinely introduced in therapies, we next tested drug combinations to prevent or delay the emergence of acquired resistance. We chose three COX-2 inhibitors and two HDAC (histone deacetylase) inhibitors that are routinely used in the veterinary practice. Drug-naive cells were treated in 9 consecutive cycles either with DOX alone, or DOX in combination with subtoxic doses (IC_80_) of SAHA (suberanilohydroxamic acid), trichostatin-A (TSA), celecoxib (CEL), firocoxib (FIR), or meloxicam (MEL). Concentrations for each drug were selected in separate cytotoxicity experiments as described in the Materials and Methods section (Appendix A). MAF was determined after every third treatment. The median time to reach MAF 0.2 (considered as the threshold of resistance), was 40, 41, 51, and 67 days for DOX + TSA (*n* = 3), DOX + SAHA (*n* = 3), DOX + MEL (*n* = 2) and DOX + FIR (*n* = 5), respectively (Figure 4a, Appendix A). Surprisingly, celecoxib completely prevented the expression of P-glycoprotein, as none of the DOX + CEL (*n* = 5) treated cultures reached the threshold of MAF = 0.2 during the 100-day-long experimental period compared to DOX (*p* = 0.0027, with hazard ratio = 27.38 and 95% CI (confidence interval) of ratio = 3.161 to 237.2) (Figure 4a,b).

The remarkable effect of CEL on DOX-induced MDR was further confirmed by drug sensitivity tests performed on both cell lines after 9 cycles of treatment. In the case of P388 cells, DOX alone resulted in a 160-fold increase in DOX resistance (IC_50_ = 0.01 µM untreated vs. IC_50_ = 1.6 µM DOX treated) while addition of CEL prevented the emergence of doxorubicin resistance (IC_50_ remained 0.03 µM) even after 9 cycles of treatment for at least 100 days (Figure 4c). In CLBL-1 cells, the DOX + CEL treatment even resulted in a decrease of DOX sensitivity (IC_50_ = 0.44 µM untreated vs. IC_50_ = 0.24 µM DOX + CEL treated), whereas DOX-treated cells were 4.5-fold resistant (IC_50_ = 1.99 µM) (Figure 4d). 

### 2.4. Celecoxib Prevents the Emergence of Drug Resistance without Inhibiting Pgp or Contributing to the Toxicity of DOX

We tested the effect of CEL on the calcein accumulation of P388/ADR cells, which overexpress Pgp as a result of continuous drug selection [33]. While P388/ADR cells show low calcein fluorescence (red) due to the Pgp-mediated efflux of calcein AM, the Pgp inhibitor verapamil restores fluorescence by blocking dye efflux (blue). In contrast, CEL (up to 160 µM) had no effect on cellular fluorescence, indicating that it does not inhibit Pgp (Figure 5a). 

Interestingly, combined treatment with DOX + CEL significantly delayed the repopulation of the flasks (8 and 22 days following treatment with DOX and DOX + CEL, respectively) (Appendix A). To rule out synergistic toxicity, the toxicity of the combined treatment was evaluated in short-term toxicity assays. Surprisingly, CEL and DOX proved to be antagonistic, rather than synergistic in both cell lines, suggesting that the ability of CEL to prevent DOX-induced MDR is not due to the increased toxicity of the combination treatment (Figure 5b,c).

To investigate the effect of CEL on Pgp expression, we cultured P388/ADR and CLBL-1 D cells in the absence or presence of CEL. Treatment with CEL decreased the MAF of both cell lines slightly more than the drug holiday (Figure 5d,e). In the case of P388/ADR cells, the MAF value changed from 0.90 (±0.08) to 0.73 (±0.1) during the 28 days of the drug holiday, whereas cells treated with CEL exhibited a MAF 0.65 (±0.1) after a 28-day-long treatment (Figure 5d). Similar results were obtained with CLBL-1 D cells (MAF 0.39 ± 0.05), exhibiting MAF = 0.30 (±0.04) following a drug holiday of 18 days, and MAF = 0.23 (±0.04) following an 18-day treatment with CEL (Figure 5e).

## 3. Discussion

Lymphoma is responsible for a significant fraction of cancer mortality in canine and human patients [1]. Novel strategies are urgently required to increase treatment efficacy in the human and veterinary clinical practice. Anthracycline-based chemotherapy protocols are initially successful, but tumor cells often become resistant [34]. Overexpression of P-glycoprotein results in multidrug resistance and an unfavorable response to therapy of canine lymphoma patients [17,35].

Several different drug resistance mechanisms were identified in cancer [36]. Malignant cells can metabolize drugs through the CYP (cytochrome P450) system [37], alter drug targets [38], increase DNA repair to overcome DNA-damaging toxic insults [39], modify apoptotic pathways to avoid cell death [40], undergo epithelial-to-mesenchymal transition [41], or use rapid epigenome modifications to reversibly adopt drug sensitive and resistant phenotypes [42]. Pgp can protect cancer cells by reducing the intracellular concentrations of drugs below a cell-killing threshold. Pgp recognizes a wide array of currently used chemotherapeutics, but the extent to which it contributes to clinical multidrug resistance has remained a matter of debate. Although the strategy of transporter inhibition could not be validated in clinical trials, mounting evidence continues to support the role of Pgp in clinical multidrug resistance in some settings [43]. Pgp contributes to therapy resistance in acute myeloid leukemia [44,45,46], ovarian and breast cancer [47,48], ependymoma [49], medulloblastoma [50], melanoma [51], and hepatocellular carcinoma [52]. Recently, we and others have shown that resistance to anthracycline-based chemotherapy is almost universally based on the overexpression of Pgp in a clinically relevant mouse model of hereditary breast cancer [53,54,55,56,57]. As a result of its broad substrate specificity, Pgp also extrudes fluorescent dyes such as calcein AM [58,59]. Clinical studies have shown that the calcein assay can be used as a quantitative, standardized, and inexpensive screening test in a routine clinical laboratory setting to detect Pgp activity and to identify patients with unfavorable therapy responses [31].

Using a functional assay, we followed the emergence of Pgp-mediated resistance in two canine patients. As expected, we found that continuous chemotherapy induced drug resistance. Surprisingly, in patient 2, a pause in the therapy resulted in a reversible reduction of resistance along with the temporary resensitization of the tumor to the treatment (Figure 1). Several studies have reported similar findings in human cancer patients, showing the beneficial effect of drug holidays on treatment efficiency. Even after therapy failure, lung cancer patients were shown to respond to EGFR (Epidermal Growth Factor Receptor) treatment following a short period of drug holiday [60,61]. Similarly, melanoma patients who discontinue BRAF (B-Raf Proto-Oncogene)-directed therapy due to progression or other causes can benefit from retreatment at a later stage [62]. The possible benefit of treatment rechallenge with chemotherapy has also been described in canine patients [63]. In the veterinary practice, drug holidays are typically inserted to reduce the toxic side effects of chemotherapy. The striking effect of the short drug holiday observed in Patient 2 prompted us to set up an in vitro assay modelling the development of clinical drug resistance and the effect of drug holiday on drug sensitivity. 

In vitro models of Pgp-mediated MDR have been typically established by the stepwise selection of cells in increasing concentrations of cytotoxic compounds [64,65]. Because of the lengthy selection procedure, MDR cells such as P388/ADR [33] express extremely high levels of P-glycoprotein, and thus do not provide a realistic model of clinical MDR. Here, we introduced a new approach for the study of acquired resistance, in which treatment with high concentrations of toxic chemotherapy is followed by recovery periods in drug-free medium. We found that repeated treatment cycles eventually led to multidrug resistant cells that express moderate, clinically relevant levels of Pgp [53,57] (Figure 2 and Figure 3). Using this protocol, we investigated the development of Pgp-mediated drug resistance and the effects of drug holiday in a mouse lymphoblastic leukemia (P388) and a canine B-cell lymphoma (CLBL-1) cell line. In line with clinical observations, we observed a rapid and reversible induction of Pgp expression in both cells. 

Following high-dose EGFR-inhibitor treatment of non-small cell lung cancer (NSCLC) cell lines, a drug-tolerant “persister” population exhibiting reversible drug resistance can be consistently detected [42]. Significantly, inhibition of the KDMA5a histone demethylase results in the selective depletion of the persister subpopulation, leading to renewed sensitivity to EGFR inhibitors. Similarly, persisters originating from T-cell acute lymphoblastic leukemia (T-ALL) rely on the chromatin modifier protein BRD4, consistent with the benefit of the BRD4 inhibitor JQ1 in primary human leukemias [25]. These findings establish the rationale for incorporating epigenetic modulators in combination therapies. To test the relevance of epigenetic regulatory mechanisms in the reversible induction of Pgp expression, we repeated our experiments in the presence of clinically used epigenetic inhibitors (SAHA, TSA). In contrast to our expectations, the inhibitors did not influence the emergence of acquired drug resistance (Figure 4a). Hematological malignancies are particularly sensitive to HDACis [66], but for our purposes the epigenetic inhibitors had to be administered at nontoxic concentrations (Appendix A), which may have resulted in suboptimal concentrations.

Another clinically feasible approach to interfere with the drug-induced overexpression of Pgp may rely on the inhibition of COX-2. Selective COX-2 inhibitors (i.e., celecoxib, firocoxib, or meloxicam) have pro-apoptotic and anti-proliferative effects in human and canine tumors [67,68,69,70]. Interestingly, Pgp was shown to mediate celecoxib-induced apoptosis by activating caspase-3 and increasing cytochrome c release from the mitochondria [71]. In addition, immunohistochemical analyses of human breast tumor specimens revealed a strong correlation between expression of COX-2 and Pgp [72]. Based on mounting evidence supporting the link between COX-2 and Pgp-mediated drug resistance [30], we tested a panel of clinically used COX-2 inhibitors in our assay.

Strikingly, addition of celecoxib completely prevented the development of MDR in both cell lines (Figure 4). Since celecoxib does not influence Pgp function (Figure 5a) or DOX toxicity (Figure 5b,c), its ability to prevent MDR is likely mediated by repressing the expression of Pgp. Although COX-2 is known to regulate Pgp expression via c-Jun phosphorylation [73] or PGE2 [74], lack of Pgp repression by firocoxib or meloxicam indicates that the effect of celecoxib is independent from the inhibition of COX-2. Inhibition of Pgp expression may be related to celecoxib’s many off-target effects, such as inhibition of cancer-associated carbonic anhydrases [75], the master kinase PDK1 [76,77], or the sarco/endoplasmic reticulum Ca^2+^-ATPase [78]. Future work will be needed to demonstrate whether this promising effect can be exploited in the human and canine cancer clinic.

## 4. Materials and Methods

### 4.1. Drugs

Doxorubicin (DOX, Sigma-Aldrich, St. Louis, MO, USA), celecoxib (CEL, Sigma-Aldrich), trichostatin-A (TSA, Tocris Bioscience, Bristol, UK), SAHA (Tocris Bioscience), and firocoxib (FIR, Sigma-Aldrich) were purchased from the manufacturers. Meloxicam (MEL, Ceva, Libourne, France) was a kind gift from Ceva Animal Health, LLC.

### 4.2. Cell Lines

The mouse leukemic P388 and its doxorubicin-selected subline P388/ADR were obtained from the National Cancer Institute’s Developmental Therapeutics Program (National Institutes of Health). The canine B-cell lymphoma CLBL-1 cell line was a kind gift from Dr. Barbara Rütgen (University of Veterinary Medicine, Vienna) [3]. P388/ADR cells were maintained in 500 nmol/L doxorubicin (Adriamycin, Sigma-Aldrich, St. Louis, MO, USA) to ensure Pgp expression. Cells were cultured in RPMI (Roswell Park Memorial Institute) media (Life Technologies, Carlsbad, CA, USA) supplemented with 10% fetal bovine serum, 5 mmol/L glutamine, and 50 units/mL penicillin and streptomycin (Life Technologies). All cell lines were cultured at 37 °C with 5% CO_2_.

### 4.3. In Vitro Cell Viability Assay

To test the cytotoxicity of mono and combined treatments, cells were seeded into 96- or 384-well tissue culture plates at 2500 (P388) or 100,000 (CLBL-1) cells/well density in 100 μL or 20 μL medium, respectively. Drug combinations were added to the plates by a Hamilton StarLet liquid handling workstation. The plates were incubated for 120 h at 37 °C with 5% CO_2_.

IC_50_ and growth inhibition (GI) values were assessed by the PrestoBlue^®^ assay (ThermoFisher, Waltham, MA, USA), according to the manufacturer’s instructions. Briefly, cells were plated in 96- or 384-well plates, treated in the given concentration range with the indicated compounds. Viability of the cells was measured spectrophotometrically using an EnSpire microplate reader (Perkin Elmer, Waltham, MA, USA). Data were normalized to untreated cells; curves were fitted by the Graph Pad Prism 8 software using the sigmoidal dose–response model. Curve fit statistics were used to determine the selected IC values. In the case of the drug combinations, GI_50_ values of ‘compound 1’ with the fixed concentrations of ‘compound 2’ (and vice versa) were paired, and plotted on an equipotent graph as GI_50_ isoboles. For each data point of the isobole, significance was calculated as the combination index (CI) [79]. Drug combinations were considered to indicate synergism (CI ≤ 0.7), moderate synergism (0.7 < CI ≤ 0.85), additive (0.85 < CI ≤ 1.2), moderate antagonism (1.2 < CI ≤ 1.45), and antagonism was defined as CI > 1.45, respectively.

### 4.4. In Vitro Model System to Study the Development of Drug Resistance

For every drug (except for doxorubicin and firocoxib), IC_80_ values (drug concentration which kills approximately 20% of the cells) were determined for both cell lines using cytotoxicity assays. In the case of doxorubicin, the concentration killing 90% of the cells (IC_10_) was used. Firocoxib, which was not toxic even at 1000 µM, was used at the concentration corresponding to the IC_80_ concentration of celecoxib. P388 cells (10^6^) were treated with 13 nM doxorubicin (DOX) for 120 h in T75 suspension flasks (Sarstedt AG & Co. KG, Nümbrecht, Germany). When used in combination, DOX was complemented with celecoxib (16 µM), firocoxib (16 µM), trichostatin-A (30 nM), or SAHA (0.4 µM). Following incubation with the drugs, surviving cells were cultured further in drug-free medium. The medium was changed every 5 days until the surviving cells reached the initial density of 10^6^/18 mL (termed as “repopulation”). Treatments were repeated multiple times; the emergence of P-glycoprotein-mediated drug-resistance was followed by the calcein assay [80]; chemosensitivity of the cells was characterized by cytotoxicity assays.

In case of CLBL-1 cells, the initial cell number was 10^7^, and the treatment lasted for 120 h. The following concentrations were used for each drug: DOX (0.3 nM), trichostatin-A (50 nM), SAHA (0.7 µM), celecoxib (26 µM), firocoxib (26 µM), and meloxicam (20 µM).

### 4.5. Immunophenotyping of CLBL-1 Cells

The following antibodies (AbD Serotec, Kidlington, UK) were used to determine the immunophenotype of the canine lymphomas by flow cytometry: CD3 FITC (clone CA17.2A12), CD5 FITC (clone YKIX322.3), CD11/18 FITC (clone YKIX490), MHC II FITC (clone YKIX334.2), CD14 PE (clone, TÜK4), CD21 PE (clone CA2.1D6), CD34 PE (clone 1H6), and CD45 APC (clone YKIX716.13). The same panel was used to verify the immunophenotype of the CLBL-1 cells (Appendix A).

### 4.6. Determination of MDR Activity Factor (MAF) with the Calcein Assay

The 250,000 cells were incubated with 0.25 mmol/L calcein AM (Dojindo Molecular Technologies, Rockville, MD, USA) in medium with or without 10 mmol/L verapamil for 10 min at 37 °C. Cells were washed with ice-cold PBS, and calcein accumulation was measured with a FACScan or FACS Calibur flow cytometer (Becton Dickinson Biosciences, San Jose, CA, USA). Dead cells were excluded based on 7-AAD (7-aminoactinomycin D (Sigma-Aldrich, St. Louis, MO, USA) permeability.

For the clinical cases, lymphoid cells were collected during surgical biopsy under general anesthesia or by fine-needle aspiration. Lymph node samples were immersed into dissociation medium containing Dulbecco’s Modified Eagle Medium (DMEM), 200 U/mL collagenase type II, and 0.6 U/mL dispase (Life Technologies, Carlsbad, CA, USA). After a 30-minute-long incubation at 37 °C, cells were separated by a 40-µM cell strainer. Isolated cells were centrifuged at 300 g. Cells were selected based on size and granularity; viable cells were gated based on 7-AAD exclusion. The 10,000 7-ADD negative cells were analyzed. The activity of Pgp was expressed as a dimensionless value using the mean fluorescence intensity measured in the presence and absence of verapamil (mean fluorescence inhibited (MFI) and noninhibited (MFNI), respectively). The MAF was determined using the following formula: MAF = (MFI − MFNI)/MFI [31,80].

### 4.7. RNA Isolation and RT-PCR

P388 cells were homogenized in TRIzol™ Reagent (Life Technologies). Total RNA was isolated using Direct-zol^®^ MiniPrep kit (Zymo Research, Irvine, CA, USA) according to the manufacturer’s instructions. To prevent DNA contamination, in-column DNAse I treatment was used. Then, 300-ng total RNA was reverse transcribed to cDNA using the Promega Reverse Transcription System Kit. Abcb1a, Abcb1b, and Actin β (Actβ) mRNA levels were quantified by TaqMan^®^ assays (ThermoFisher), using the StepOne™ Real-Time PCR System (Life Technologies). The mRNA fold changes were determined by the 2^−ΔΔCt^ method.

### 4.8. Canine B-Cell Lymphoma Case Studies

Two canine patients diagnosed with B-cell lymphoma at the Veterinary Hematology and Oncology Clinic (Budapest) were treated according to the modified CHOP protocol consisting of doxorubicin (Adriamycin injection, Pharmacia & Upjohn S.p.A. Co., Milan, Italy) 30 mg/m^2^: week 2, 11, 20; vincristi (Vincristine liquid injection, Gedeon Richter Co., Budapest, Hungary) 0.75 mg/m^2^: week 1, 3–10, and 12–19; cyclophosphamide (Endoxan injection, Baxter Co., Deerfield, IL, USA) 250 mg/m^2^: week 5, 8, 14, 17; prednisolone (Prednisolone tablet, Gedeon Richter Co., Budapest, Hungary): 2 mg/kg BW (body weight) week 1, perorally (po) daily, once a day (SID), 1.5 mg/kg BW week 2 po daily, SID, 1 mg/kg BW week 3 po daily, SID, 0.5 mg/kg BW week 4, po daily, SID [81].

Case 1: German shepherd, male, 3.5 years old. Large cell immunoblastic lymphoma was diagnosed in stage V (substage b). Case 2: Cocker spaniel, female, 6.5 years old. Diffuse large B-cell lymphoma was diagnosed in stage IV (substage a). All dogs were staged and substaged according to the scheme established by the World Health Organization (WHO) [82].

Initial diagnosis was made by chest X-ray, abdominal ultrasonography, complete blood count, and routine plasma clinical chemistry analyses together with right prescapular lymph node excision and bone marrow aspiration cytology based on the informed consent of the owner (Appendix A). Histopathology and immunophenotyping of the tumor cells were determined by FACS analysis of the excised lymph node. At the time of diagnosis, during treatment, and at the end of chemotherapy, Pgp-mediated drug resistance was monitored by the calcein assay and the immunophenotype of tumor cells was characterized by flow cytometry.

### 4.9. Statistical Analysis

Statistical analyses were performed using the GraphPad Prism version 8.0.0 for Windows, GraphPad Software (San Diego, CA, USA). One-way or two-way ANOVA followed by Tukey’s multiple test was used for comparisons between treatment groups and MAF values for each cell line. The difference between Kaplan–Meier survival curves was determined by log-rank test. The *p* < 0.05 was considered as statistically significant.

## 5. Conclusions

In conclusion, this work demonstrates that celecoxib effectively blocks the emergence of multidrug resistance by preventing the doxorubicin-induced upregulation of P-glycoprotein. We demonstrated that doxorubicin treatment induces MDR in lymphoma cell lines, while addition of celecoxib inhibits the development of resistance. Since celecoxib is not a Pgp inhibitor and the combination of the two drugs did not increase toxicity, we conclude that celecoxib prevents the emergence of MDR by interfering with Pgp expression. Our results show that the drug holiday effect can be mimicked in vitro with a clinically used COX-2 inhibitor, offering a novel strategy to prolong response to therapy and to delay or prevent the development of drug resistance. Based on these results a randomized double-blind controlled study was initiated in the Veterinary Hematology and Oncology Clinic in Hungary.

## Figures and Tables

**Figure 1 cancers-12-01117-f001:**
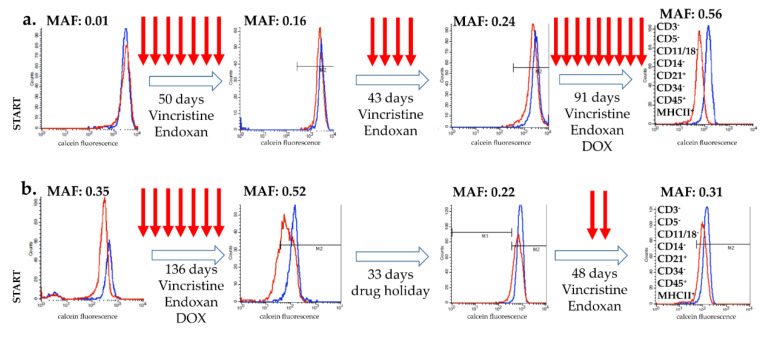
Changes in P-glycoprotein function during chemotherapy in two B-cell lymphoma canine patients. P-glycoprotein activity, as determined by the calcein assay (MAF values), was monitored during therapy (treatments are indicated by arrows). Flow cytometry histograms show the calcein fluorescence of cancer cells incubated with (blue) and without (red) the Pgp inhibitor verapamil. (**a**) Patient 1 showed the typical kinetics of acquired multidrug resistance. (**b**) In the case of Patient 2, MDR was reversed by a 33-day-long (unplanned) drug holiday. The immunophenotype of the lymphoma remained identical during the treatment in both case (results show data of the final measurement).

**Figure 2 cancers-12-01117-f002:**
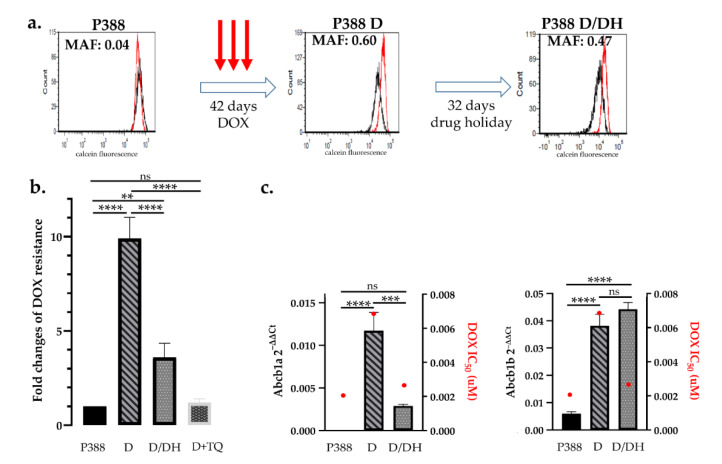
Effect of doxorubicin treatment and drug holiday on mouse P388 lymphoblastic leukemia cells. (**a**) Parental P388 cells were treated with 13 nM DOX. After 3 cycles of DOX treatment (42 days) P388 D cells showed a significant increase in P-glycoprotein activity (MAF 0.6 vs. MAF 0.04), which was significantly reduced after a 32-day-long drug holiday (MAF 0.47). Flow cytometry histograms show the results of the calcein assay of cells assayed in the presence (red) or absence (black) of the Pgp inhibitor verapamil. (**b**) Changes of doxorubicin sensitivity as a result of drug treatment and drug holiday. Sequential DOX treatments of P388 cells resulted in a 9.9-fold increase of doxorubicin tolerance (P388 D), which was significantly reduced following a drug holiday (P388 D/DH). Resistance of P388 D cells was abrogated in the presence of tariquidar (P388 D + TQ) (**c**) Abcb1a and b mRNA expression and DOX IC_50_ values (red dots) of P388 parental cells (P388) after DOX treatment (D) and following drug holiday (D/DH). Statistical analysis was performed on mRNA samples, ** *p* < 0.01, *** *p* < 0.001, **** *p* < 0.0001.

**Figure 3 cancers-12-01117-f003:**
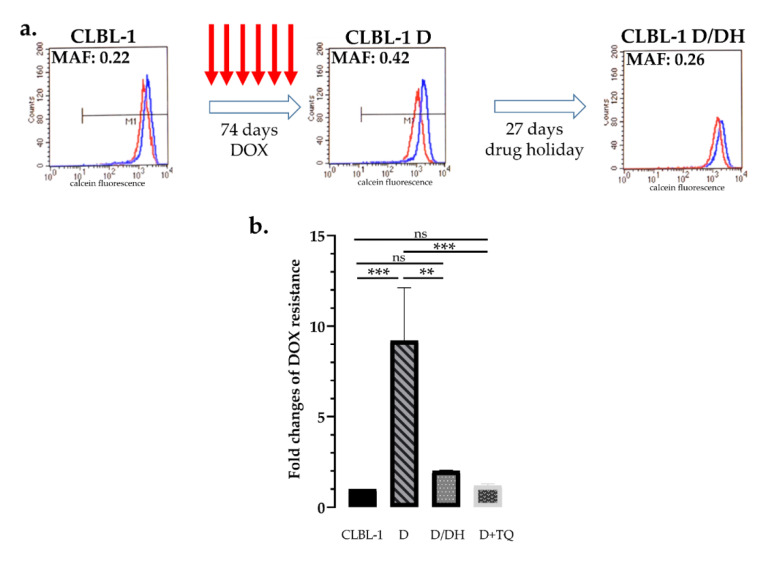
Effect of doxorubicin treatment and drug holiday on canine CLBL-1 B-cell lymphoma cells. (**a**) After 6 cycles of DOX treatment (74 days) parental CLBL-1 cells showed a significant increase in P-glycoprotein activity (MAF 0.42 vs. 0.22), which was significantly reduced after a 27-day-long drug holiday (MAF 0.26). Flow cytometry histograms show the results of the calcein assay of cells assayed with (blue) or without (red) the Pgp inhibitor verapamil. (**b**) Changes of doxorubicin sensitivity as a result of drug treatment and drug holiday. Sequential DOX treatments of CLBL-1 cells resulted in a 9.2-fold increase of doxorubicin tolerance, which was significantly reduced following the period of drug holiday. Resistance of CLBL-1 DOX cells was abrogated in the presence of tariquidar (D + TQ). ** *p* < 0.01; *** *p* < 0.001; ns: not significant.

**Figure 4 cancers-12-01117-f004:**
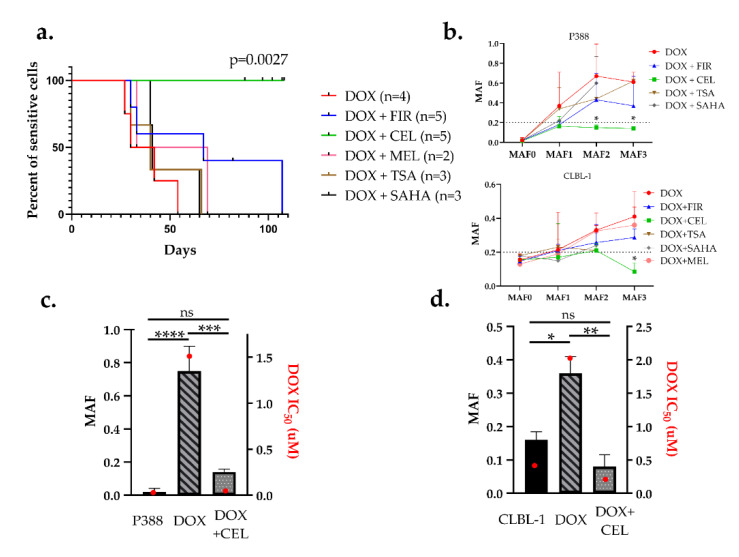
Celecoxib prevented the emergence of Pgp-mediated drug resistance in both P388 and CLBL-1 cells. (**a**) Kaplan–Meier curves of cell cultures treated with the indicated combinations. Cells were considered resistant at MAF ≥ 0.2. DOX (red) in combination with firocoxib (blue), celecoxib (green), meloxicam (pink), trichostatin-A (brown), and SAHA (black) were treated in 9 sequential treatment cycles. (**b**) Quantitative evaluation of multidrug resistance during the course of various treatments in P388 and CLBL-1 cells. MAF was measured after every third treatment cycle (MAF1–3). (**c**,**d**) Relation of MAF and drug sensitivity. MAF (patterned bars) and DOX IC_50_ values (red squares) of parental (sensitive), DOX-treated, and DOX + CEL-treated (**c**) P388 cells, and (**d**) CLBL-1 cells, measured after 9 sequential treatment cycles (at around day 100). Statistical analysis was performed on MAF values, * *p* < 0.05, ** *p* < 0.01, *** *p* < 0.001, **** *p* < 0.0001, ns: not significant.

**Figure 5 cancers-12-01117-f005:**
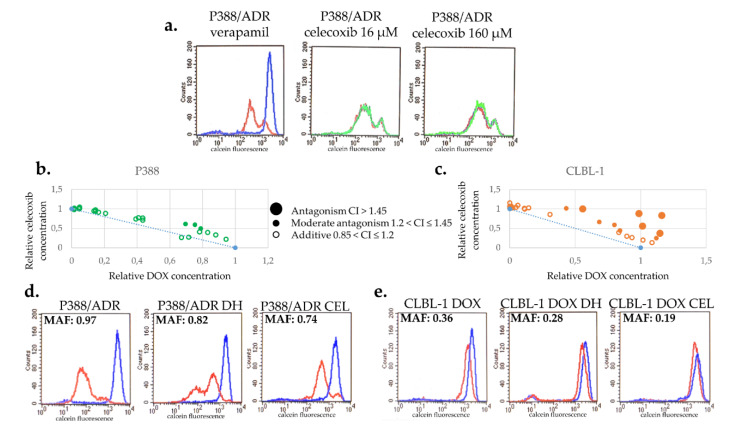
Celecoxib is not a Pgp inhibitor and does not increase the toxicity of DOX. (**a**) In contrast to verapamil (blue), a known Pgp inhibitor, 16–160 µM CEL (green) does not inhibit the function of Pgp. (**b**,**c**) CEL does not show synergistic toxicity with DOX in P388 (**b**) or in CLBL-1 (**c**) cells. (**d**,**e**) Effect of CEL treatment on MDR in Pgp-expressing P388/ADR (**d**) and CLBL-1 D (**e**). CEL treatment resulted in a more pronounced reduction of MAF in Pgp-expressing P388/ADR (**d**) and CLBL-1 D (**e**) cells than drug holiday (DH).

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
