# Peer review of "Celecoxib Prevents Doxorubicin-Induced Multidrug Resistance in Canine and Mouse Lymphoma Cell Lines"

_cancers, 2020, doi:10.3390/cancers12051117_

Round 1
Reviewer 1 Report
The study "Celecoxib prevents doxorubicin-induced multidrug resistance in canine and mouse lymphoma cell lines" by Karai and colleagues describes the potential treatment of multidrug resistance (MDR) by targeting P-glycoprotein (Pgp) upregulation. The authors identified that Pgp, which extrudes chemotherapeutic agents from cancer cells, increases in response to chemotherapy regimens. However, “drug holidays” (intervals between chemotherapy) can decrease Pgp and promote drug sensitivity. Moreover, the authors identified that celecoxib, a COX-2 inhibitor, prevents the development of Pgp-mediated drug resistance in vitro. This work suggests the efficacy of using celecoxib for preventing MDR and thus improving chemotherapy. This study is very well done and the experiments are both comprehensive as well as logical. The use of canine patients and cell lines provide strong evidence for the authors claims. The major issue with the manuscript is the lack of statistics. However, with modifications, this paper has the potential to modify clinical practice.
Major Comments:
- There are no statistics in the paper. All the data must be analyzed by One Way ANOVA; Kaplan-Meier estimator (for survival data); or 2 WAY ANOVA (for multidrug resistance during the course of various treatments; Figure 4). Please run a regression analysis for Figure 5b-c. Include all the statistical analyses in the Methods section.
- Along these same lines, please include the ± Sdev for all the MAF data (which presumably was run more than 1x).
Author Response
First of all, we would like to thank the Reviewer for the thorough inspection of our manuscript and the helpful comments to further improve our work!
The recommended statistical analyses were performed. We included the results in the amended manuscript, and we also updated the figures to indicate statistical significance. We added the ± SD to all MAF values, except for the patient samples, which were measured only once at each timepoint. The following text was added to the Materials and Methods section:
“5.9 Statistical Analysis
Statistical analyses were performed using the GraphPad Prism version 8.0.0 for Windows, GraphPad Software (San Diego, California USA). One-way or two-way ANOVA followed by Tukey’s multiple test was used for comparisons between treatment groups and MAF values for each cell line. The difference between Kaplan-Meier survival curves was determined by Log-Rank test. p < 0.05 was considered as statistically significant.”
Regarding the drug synergism plots (Fig5b-c) we would respectfully argue against the use of regression analysis. Drug combination studies are rarely, if ever, analyzed with regression analysis. In a seminal paper published in 2006, Ting-Chao Chou analyzed different methods used to interpret drug synergism studies. Based on this widely cited study, the best practice is based on the evaluation of Combination Index (CI) values (Chou TC: Theoretical basis, experimental design, and computerized simulation of synergism and antagonism in drug combination studies, cited more than 3400 times!). In line with these recommendations, GI50 values of doxorubicin with the fixed concentrations of celecoxib (and vice versa) were paired, and plotted on an equipotent graph as GI50 isoboles. For each data point of the isobole, significance was calculated as the CI.
Reviewer 2 Report
This manuscript reports celecoxib prevents doxorubicin-induced multidrug resistance, not directly but through a calcein assay process. In general, the organization of this manuscript is very focused. Readability is good as it appears. Schemes, tables, as well as references allow to get a good understanding of the study.
Major comments:
- Cellular MDR mechanisms are modified to escape drug effect include elevated DNA repair, increased drug metabolism, altered apoptotic pathways to bypass drug targets, loss or change of drug target proteins, and increased efflux of anticancer drugs (e.g. altered activities of membrane transporters such as ATP-binding cassette transporters). The authors claimed that “celecoxib effectively blocks the emergence of multidrug resistance by preventing the doxorubicin-induced upregulation of P-glycoprotein”. The calcein assay (multidrug resistance activity factor) was used to detect both P-glycoprotein and multidrug resistance-associated protein activities. More information or work is needed for the conclusion. Please fill some more background, relationship and evidences.
- This work lacks the in vivo data to show the importance of celecoxib prevents doxorubicin-induced multidrug resistance in B-cell lymphoma.
Minor comments:
- Please put a space in between the number and the unit such as μM, °C, and nM. The authors should pay attention on the subscript of indices and insertion of space where necessary, check throughout the manuscript.
- Figure 2c, please correct a typo. It should be Abcb1b?
Author Response
Major comments:
Cellular MDR mechanisms are modified to escape drug effect include elevated DNA repair, increased drug metabolism, altered apoptotic pathways to bypass drug targets, loss or change of drug target proteins, and increased efflux of anticancer drugs (e.g. altered activities of membrane transporters such as ATP-binding cassette transporters). The authors claimed that “celecoxib effectively blocks the emergence of multidrug resistance by preventing the doxorubicin-induced upregulation of P-glycoprotein”. The calcein assay (multidrug resistance activity factor) was used to detect both P-glycoprotein and multidrug resistance-associated protein activities. More information or work is needed for the conclusion. Please fill some more background, relationship and evidences.
We are grateful for the Reviewer’s valuable comments, which helped us to improve the quality of our work.
According to the Reviewer’s suggestion we inserted a new paragraph into the Discussion to better establish the scientific framework of our study. We provide more information on the relevance of Pgp in drug resistance, and the value of the Calcein AM assay in monitoring Pgp function (page 7, 8):
“Several different drug resistance mechanisms were identified in cancer [35]. Malignant cells can metabolize drugs through the CYP system [36], alter drug targets [37], increase DNA repair to overcome DNA damaging toxic insults [38], modify apoptotic pathways to avoid cell death [39], undergo epithelial-to-mesenchymal transition [40] or use rapid epigenome modifications to reversibly adopt drug sensitive and resistant phenotypes [41]. Pgp can protect cancer cells by reducing the intracellular concentrations of drugs below a cell-killing threshold. Pgp recognizes a wide array of currently used chemotherapeutics, but the extent to which it contributes to clinical multidrug resistance has remained a matter of debate. Although the strategy of transporter inhibition could not be validated in clinical trials, mounting evidence continues to support the role of Pgp in clinical multidrug resistance in some settings [42]. Pgp contributes to therapy resistance in acute myeloid leukemia [43–45], ovarian and breast cancer [46, 47], ependymoma [48], medulloblastoma [49], melanoma [50] and hepatocellular carcinoma [51]. Recently, we and others have shown that resistance to anthracycline-based chemotherapy is almost universally based on the overexpression of Pgp in a clinically relevant mouse model of hereditary breast cancer [52–56]. As a result of its broad substrate specificity, Pgp also extrudes fluorescent dyes such as Calcein AM [57, 58]. Clinical studies have shown that the calcein assay can be used as a quantitative, standardized and inexpensive screening test in a routine clinical laboratory setting to detect Pgp activity, and to identify patients with unfavourable therapy responses [30].”
According to the Cancers journal guidelines the Conclusion should be only a short and concise section, however, in compliance with the reviewer’s suggestion, we supplemented it with the following sentences:
“We demonstrated that doxorubicin treatment induces MDR in lymphoma cell lines, while addition of celecoxib inhibits the development of resistance. Since celecoxib is not a Pgp inhibitor and the combination of the two drugs did not increase toxicity, we conclude that celecoxib prevents the emergence of MDR by interfering with Pgp expression.”
This work lacks the in vivo data to show the importance of celecoxib prevents doxorubicin-induced multidrug resistance in B-cell lymphoma.
We couldn’t agree more – the in vivo data will be very interesting! Based on the promising results obtained in the lab, we initiated a randomized double-blind study to explore the effect of celecoxib supplementation of the CHOP treatment protocol in canine cancer patients. Three treatment groups will be established, receiving cyclophosphamide + doxorubicin + vincristine + prednisolone), CHOP + celecoxib or the CHO protocol (cyclophosphamide + doxorubicin + vincristine) + celecoxib. Each group will consist of at least 30 canine patients. In addition to routine clinical parameters, we will monitor the MDR Activity Factor in all patients. This study will prove whether celecoxib can influence the outcome of chemotherapy by delaying or preventing the emergence of Pgp-mediated resistance. Our clinical study has recently been approved by the national authorities, and now we are in the recruitment stage. The planned duration of the trial is 2 years.
Minor comments:
Please put a space in between the number and the unit such as μM, °C, and nM. The authors should pay attention on the subscript of indices and insertion of space where necessary, check throughout the manuscript.
Figure 2c, please correct a typo. It should be Abcb1b?
The required spaces were added, and the typos were corrected.
Reviewer 3 Report
The manuscript entitled „Celecoxib prevents doxorubicin-induced multidrug resistance in canine and mouse lymphoma cell lines” by Edina Karai et al. describes interesting studies on a reversal action towards cancer multidrug resistance associated with ABCB1 transporters, caused by either “drug holiday” or selected COX-2 inhibiting drugs. The presented approach is innovative and seems to be in great importance in order to improve not satisfying therapy of tumor, significantly limited by various mechanisms of multidrug resistance, in particular those involving ABC drug efflux pumps, where ABCB1 (Pgp) is a predominant one. The scientific attractiveness of this study, is contained in the use of canine B-lymphoma cell model as B-lymphocytes are more often than T associated with lymphoma cancer and this, together with a high analogy of canine to human B-lymphoma, seems to be very promising for future therapy. Furthermore, the scientific level of the assays performed, involving RT-PCR and spectrophotometrically-supported molecular biology methods, is high enough for a reputable journal. Thus, I recommend this work to be published in MDPI Cancers, but several minor points should be addressed before, as follows:
1. Line 71 it is an abbreviation „HDACis”, while the same inhibitors are shortened as „HDAC” in the line 78. It should be homogenized within the manuscript.
2. English style needs careful modifications, i.e. some fragments are described in past simple while other corresponding in present simple, e.g. at the end of Introduction, lines 76-79 are in past simple, followed by present simple in the lines 79-82.
3. In the Line 41, there is explained CHOP protocol, then, in line 87, Authors refer to COPA protocol not explained directly. According to the experimental data provided by Authors the drug components are the same. Are CHOP and COPA the same protocols? The abbreviation COPA should be explained, when appears first time in the text.
4. Graphical part is not sufficiently prepared to provide appropriate data Authors refer to. Descriptions at graphs in Figures 1-5 need a good microscope to be decrypted. Furthermore Authors placed three Tables and four Figures in Supplementary material , often referring to them, while these data are not available for a reader as the provided link is invalid when click on (at least for me). This “unhealthy” space saving should be modified to make the results fully legible for readers.
Author Response
The manuscript entitled „Celecoxib prevents doxorubicin-induced multidrug resistance in canine and mouse lymphoma cell lines” by Edina Karai et al. describes interesting studies on a reversal action towards cancer multidrug resistance associated with ABCB1 transporters, caused by either “drug holiday” or selected COX-2 inhibiting drugs. The presented approach is innovative and seems to be in great importance in order to improve not satisfying therapy of tumor, significantly limited by various mechanisms of multidrug resistance, in particular those involving ABC drug efflux pumps, where ABCB1 (Pgp) is a predominant one. The scientific attractiveness of this study, is contained in the use of canine B-lymphoma cell model as B-lymphocytes are more often than T associated with lymphoma cancer and this, together with a high analogy of canine to human B-lymphoma, seems to be very promising for future therapy. Furthermore, the scientific level of the assays performed, involving RT-PCR and spectrophotometrically-supported molecular biology methods, is high enough for a reputable journal. Thus, I recommend this work to be published in MDPI Cancers, but several minor points should be addressed before, as follows:
The authors would like to thank the reviewer’s support and insightful comments, which made this manuscript notably better.
- Line 71 it is an abbreviation „HDACis”, while the same inhibitors are shortened as „HDAC” in the line 78. It should be homogenized within the manuscript.
The abbreviation of HDAC inhibitors is now uniformly changed to HDACi throughout the manuscript.
- English style needs careful modifications, i.e. some fragments are described in past simple while other corresponding in present simple, e.g. at the end of Introduction, lines 76-79 are in past simple, followed by present simple in the lines 79-82.
The paper was carefully revised to improve the grammar and readability. However, we feel that in the last sentence of the introduction (lines 79-82), the use of the simple present (“here we show”) is appropriate, as the statement refers to the current paper. “We showed” would refer to a previous paper. In any case, we leave the decision to the Editor’s judgement call.
- In the Line 41, there is explained CHOP protocol, then, in line 87, Authors refer to COPA protocol not explained directly. According to the experimental data provided by Authors the drug components are the same. Are CHOP and COPA the same protocols? The abbreviation COPA should be explained, when appears first time in the text.
We would like to thank for the reviewer for spotting this mistake in the manuscript! The CHOP (cyclophosphamide (C), doxorubicin (H, hydroxydaunorubicin), vincristine (O, Oncovin), and prednisone (P)) and the COPA (cyclophosphamide (C), vincristine (O, Oncovin), prednisone (P) and doxorubicin (A, adriamycin) protocols are the same, however “CHOP” is the officially used name for this treatment regimen. In the amended version we only use the CHOP abbreviation.
- Graphical part is not sufficiently prepared to provide appropriate data Authors refer to. Descriptions at graphs in Figures 1-5 need a good microscope to be decrypted. Furthermore Authors placed three Tables and four Figures in Supplementary material , often referring to them, while these data are not available for a reader as the provided link is invalid when click on (at least for me). This “unhealthy” space saving should be modified to make the results fully legible for readers.
According to the reviewer’s suggestion we increased the size of the figures and the related legends. Surprisingly, the Supplementary Figure file, which we’ve uploaded according to the journal’s guidelines, was not provided to the reviewers for an unknown reason. Hopefully, in this round of review, all of the reviewers can access this document also. We agree with the reviewer that the data shown in the Supplementary Material are important, however the detailed, weekly description of the CHOP treatment protocol and all the cytotoxicity curves measured for 6 drugs on 2 cell lines in vitro among others, would make this manuscript unnecessarily long. We will draw the journal’s attention to this issue and surely, they are going to make this document available for all the readers.